# Coverage-Driven KV Cache Eviction for Efficient and Improved Inference of LLM

## Abstract

Large language models (LLMs) excel at complex tasks like question answering and summarization, thanks to their ability to handle long-context inputs. However, deploying LLMs is costly, not only due to the high computational demands of quadratic complexity of self-attention and auto-regressive generation, but also because of the significant memory overhead required for storing the key-value (KV) cache during inference. To reduce the memory cost, existing KV-cache eviction strategies leverage the sparsity in attention to selectively store a subset of tokens. While reducing the memory footprint, such approaches show a considerable drop in performance, especially in tasks that require long-context reasoning. We identify that the drop in performance is linked to a reduction in the coverage of unique tokens. Additionally, we theoretically show that reduced coverage limits the mutual information between inputs and outputs, thereby impairing predictive accuracy. To this end, we introduce K-VEC, a novel coverage-aware KV-cache eviction strategy that prioritizes token coverage while evicting tokens in the cache. K-VEC introduces a cross-head and a cross-layer coverage module to enhance token retention across attention heads and model layers, mitigating performance degradation caused by low coverage. Evaluated on 16 LongBench subsets, K-VEC exhibit up to 10.35 points improvement over the existing methods under the same eviction rate and memory constraint. Comprehensive evaluations validate the effectiveness of our approach and demonstrate its potential for efficient LLM deployment in resource-constrained settings.

## 1 Introduction

Large language models (LLMs) have demonstrated remarkable performance in tasks such as question answering, retrieval-augmented generation, summarization, and dialogue systems, largely due to their ability to process and reason over long-context inputs. However, deploying LLMs remains computationally expensive due to their auto-regressive generation and the quadratic complexity of self-attention, which scales with input length. Additionally, the key-value (KV) cache, used to store intermediate attention states and avoid recomputation during inference, imposes substantial memory overhead. Its size grows with both sequence length and model depth, limiting scalability in resource-constrained or real-time settings. To address this, recent works such as H2O (Zhang et al., 2024b), PyramidKV (Zhang et al., 2024a), and SnapKV (Li et al., 2024) have proposed eviction strategies that leverage attention sparsity to selectively store only a subset of the KV cache. Besides reducing the memory usage, the reduced sequence length also improves the computational cost during inference.

While existing KV-cache eviction strategies (Li et al., 2024; Zhang et al., 2024b;a; Feng et al., 2024) perform well at moderate eviction rates, their effectiveness diminishes at higher rates, particularly for tasks requiring long-context reasoning. Our preliminary analysis, illustrated in Figure 1, reveals a strong correlation between performance degradation and reduced coverage, where coverage denotes the number of unique tokens retained in the KV cache (at least once, across the heads and layers). We further provide a theoretical basis for these findings using an information bottleneck framework, demonstrating that lower coverage limits the mutual information between input and output, thereby degrading the model's predictive performance.

In this work, we propose K-VEC (KV Cache Eviction with Coverage), a coverage-aware KV cache eviction strategy designed to enhance the diversity of cached tokens. K-VEC addresses a key limitation of existing methods: eviction scores across attention heads and layers frequently overlap, resulting in the eviction of similar tokens and a subsequent reduction in overall coverage. To counter this, K-VEC introduces two novel components: **cross-head coverage** and **cross-layer coverage**. Specifically, the cross-head coverage module increases coverage by adjusting the attention span of a few strategically selected heads to focus on diverse tokens. On the other hand, the cross-layer coverage module improves coverage by encouraging the selection of globally important tokens that are not selected in earlier layers. Together, these components ensure that the most important tokens are retained in the KV cache, enabling the LLM to generate the desired response.

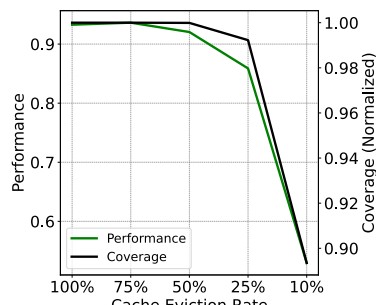

Figure 1: Correlation between reduced coverage and performance (F1 score) degradation for Llama-3.1-8B-Instruct using SnapKV on TriviaQA, highlighting the importance of coverage.

We evaluate K-VEC on the established protocol from prior works (Li et al., 2024; Feng et al., 2024) and report results on 16 subsets of the LongBench dataset, covering single-document QA (Kočiskỳ et al., 2018; Dasigi et al., 2021), multi-document QA (Yang et al., 2018; Ho et al., 2020; Trivedi et al., 2022), summarization (Zhong et al., 2021; Fabbri et al., 2019), few-shot learning (Li & Roth, 2002; Gliwa et al., 2019; Joshi et al., 2017), synthetic tasks, and code (Guo et al., 2023; Liu et al., 2023). Evaluations are conducted across various KV cache budget sizes (128, 256, 512, and 1024 tokens). K-VEC demonstrates significant improvements over existing methods, with particularly pronounced gains at lower cache budgets (e.g., 128 tokens), showing up to 10.35 points improvement over the existing methods under the same memory constraint. The average improvement on the 16 datasets is 1.61 points. We provide comprehensive ablation and sensitivity analysis to validate the effectiveness of each proposed module. Overall, our contributions are as follows:

- We identify reduced token coverage as the primary cause of performance degradation in existing KV cache eviction methods and provide a theoretical foundation for these empirical observations.

- We introduce K-VEC, a novel KV cache eviction strategy that enhances both head-wise and layer-wise token coverage.

- Extensive experiments demonstrate that K-VEC outperforms existing KV cache eviction methods while maintaining comparable computational efficiency. To foster quick reproduction and further development in this area, we will make the code publicly available upon acceptance.

## 2 Related Work

The substantial memory demands of storing KV caches during long-sequence inference pose significant challenges for LLMs, leading to high memory consumption and I/O latency (Wang & Chen, 2023). A prominent approach to mitigating this issue involves cache size reduction through eviction, using various scoring functions to assess token importance. There are two broad groups of eviction policies: sliding window eviction and top-k eviction. Sliding window methods, such as StreamingLLM (Xiao et al., 2023), preserve the initial tokens and those within a fixed window, discarding others (Beltagy et al., 2020; Han et al., 2024; Xiao et al., 2023). While straightforward, this non-selective approach often degrades output quality in long-context scenarios. In contrast, top-k eviction methods (Ge et al., 2023; Liu et al., 2024; Ren & Zhu, 2024; Zhang et al., 2024b; Yang et al., 2024; Zhang et al., 2024a; Li et al., 2024) retain the $k$ most important tokens based on attention scores to maintain post-eviction performance. For example, FastGen (Ge et al., 2023) combines retention of special tokens, punctuation, recent tokens, and top-k selections, adapting to head-specific attention patterns. H2O (Zhang et al., 2024b) selects critical tokens using query states across all positions. More advanced methods like SnapKV (Li et al., 2024) and Pyramid (Yang et al., 2024; Zhang et al., 2024a) enhance efficiency by prioritizing query states within a local observation window. Additionally, Pyramid introduced a layer-wise adaptive budget allocation strategy. Overall, most existing approaches

allocate the cache budget uniformly across attention heads. More recently, AdaKV introduced a head-wise adaptive budget allocation strategy that can be used as a plug-and-play enhancement to improve the performance of existing methods. Nonetheless, all current methods compute eviction scores independently across heads and layers, lacking any inherent mechanism to encourage diverse token coverage. As a result, similar tokens are often evicted across both heads and layers, leading to a reduction in overall coverage.

# 3 Coverage Hypothesis for KV-Cache Eviction

## 3.1 Preliminary Studies

In this section, we investigate the cause of coverage loss in existing KV-cache eviction methods, such as SnapKV (Li et al., 2024). Specifically, we analyze the eviction scores used by SnapKV across attention heads and layers. Figure 2 illustrates an example of these scores over the input tokens, where tokens with lower scores are removed. As shown in the figure, there is a pronounced bias toward the end of the input prompt, favouring the eviction of earlier tokens. This skewness results in a disproportionate retention of tokens from the end of the prompt, a pattern that remains consistent across heads and layers. Consequently, similar tokens, primarily from the end of the input prompt, are retained across heads and layers, leading to a significant reduction in overall token coverage.

The impact of this phenomenon is illustrated in Figure 3, using an example from a multiple-choice question-answering task with four options. In the figure, darker colours indicate tokens retained by a higher number of layers (up to a maximum of 32), while uncoloured tokens are ignored by all layers. Tokens that are attended by no layers represent a loss of information, which can degrade performance, especially when the tokens contain important information regarding the query. In this example, the correct answer (the first option) receives minimal attention, while the model heavily focuses on the final few tokens of the prompt across all layers, highlighting a strong recency bias in SnapKV's eviction strategy. This bias reduces coverage of critical tokens and contributes to performance degradation in long-context tasks. In the next sub-section, we provide a coverage hypothesis, a theoretical basis for the relation between coverage and performance.

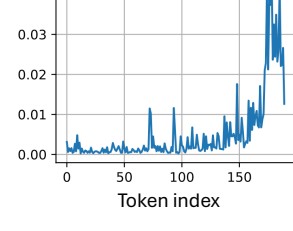

Figure 2: Windowed attention scores in SnapKV are skewed toward the end of the sequence. The visualization is based on the first head of the first layer; the same pattern persists across layers and heads.

## 3.2 Coverage Hypothesis

Let $X$ denote an input sequence and $Y$ the corresponding output sequence. A transformer model computes hidden representations at each layer via attention mechanisms operating on keys $K$ and values $V$. Given a cache, storing processed tokens, $\mathcal{T} = \{t_1, t_2, \ldots, t_N\}$, where $N$ is the total number of tokens selected across all attention heads and layers, we define *coverage $C$* as:

$$C = |\{\text{unique tokens in } \mathcal{T}\}|,$$

with $\mathcal{T}_{\text{eff}}$ denoting the effective token set after eviction, where $|\mathcal{T}_{\text{eff}}| = C$.

Let $I(X; Y)$ represent the mutual information between input and output. We analyze how cache coverage constrains this quantity through the lens of information bottleneck theory.

**The Information Bottleneck via Cache Eviction.** The transformer's intermediate representation $\mathcal{T}$ mediates information flow through the cache. By the Data Processing Inequality (DPI):

$$I(X; Y) \leq I(\mathcal{T}; Y). \tag{1}$$

If $X \to \mathcal{T} \to Y$ forms a Markov chain, $\mathcal{T}$'s capacity fundamentally limits prediction performance.

**Bounding $I(\mathcal{T}; X)$ via Coverage.** Assuming each unique token provides at most $R$ bits of information:

$$I(\mathcal{T}; X) \leq R \cdot C. \tag{2}$$

Figure 3: Coverage of tokens over input prompt for existing method. The existing method's token coverage skews heavily toward the end of the input, despite the correct answer appearing first in this example.

In practice, however, large vocabularies often contain redundancy, where additional unique tokens yield diminishing information gains. To capture this effect, we introduce a tighter logarithmic bound:

$$I(\mathcal{T}; X) \leq \alpha \log C, \quad \alpha > 0. \tag{3}$$

**Relating Mutual Information to Performance.** Let $P$ be a performance metric (e.g., negative log-likelihood) with:

$$P \geq \phi(I(X; Y)), \tag{4}$$

where $\phi(\cdot)$ is monotonically decreasing. Combining with DPI:

$$P \geq \phi\left(\min\{I(\mathcal{T}; X), I(\mathcal{T}; Y)\}\right) \tag{5}$$
$$\geq \phi(\alpha \log C). \tag{6}$$

To achieve target performance $P_0$, coverage must satisfy:

$$C \geq \exp\left(\frac{1}{\alpha} \phi^{-1}(P_0)\right). \tag{7}$$

This bound relies on the assumption that the token contributions are additive (no contextual dependencies), and all tokens as equally informative, though some may carry more task-relevant signal. Our goal is to establish the relationship between the coverage and performance drop, rather than to quantify the exact decline or relate it to the specific importance of individual tokens. This assumption allows for a concise and transparent theoretical derivation aligned with our objective. Under this assumption, the analysis reveals an irreducible trade-off: a drop in coverage during eviction ($C \downarrow$) forces $I(X; Y) \downarrow$, explaining the empirical performance collapse.

## 4 KV Cache Eviction with Coverage

To maximize token coverage during KV cache eviction, we propose **K-VEC** (*KV Cache Eviction with Coverage*), a strategy that operates on two levels: *across attention heads* and *across transformer layers*, through our *cross-head coverage* and *cross-layer coverage* modules. Below, we detail our proposed solution designed to increase the overall coverage of evicted tokens.

### 4.1 Cross-Head Coverage

As discussed in the preliminaries, the eviction policy often selects overlapping tokens across heads, resulting in reduced coverage across the heads of a layer. To this end, we introduce the cross-head coverage module to expand the coverage across heads. Existing eviction policy (e.g., SnapKV) typically computes token importance at a certain head of layer $\ell$ using a windowed attention score. Specifically, with attention scores $A \in \mathbb{R}^{L \times H \times T \times T}$ (number of layers L, number of heads $H$, sequence length $T$), the eviction scores are computed over an observation window of length $O$ as:

$$P_{\ell,h,t} = \frac{1}{O} \sum_{q=T-O+1}^{T} A_{\ell,h,q,t}. \tag{8}$$

This formulation captures the average attention over the most recent $O$ (e.g., 16) tokens, which serves as a proxy for estimating token importance in the eviction decision. Given a cache budget of $B$ tokens and $P \in \mathbb{R}^{H \times T}$, each head $h$ selects the top-$B$ tokens

$$\mathcal{T}_h = \{t \mid P_{\ell,h,t} \text{ among top } B \text{ of } \{P_{\ell,h,1}, \ldots, P_{\ell,h,T}\}\}, \tag{9}$$

from the $T$ input tokens. Subsequently, the key-value corresponding to the selected $B$ tokens is stored in the cache for subsequent decoding steps.

Our proposed cross-head coverage module utilizes the property that the important score is conditioned on the observation window $O$, which can be leveraged to encourage the selection of more diverse yet important tokens across heads. Specifically, by increasing the observation window to $O'$ ($O' > O$) for calculating the eviction score (using Eq. 8), we encourage the eviction score to focus on a broader context. However, our goal is not only to maximize coverage but also to ensure that the selected tokens are contextually important. To strike this balance, we selectively update the priority scores for only a subset of $\delta$ heads. These heads are chosen based on how focused their eviction score is. For example, a head with uniform scores over the tokens lacks focus and is less effective for optimal performance. We use the standard deviation of eviction scores over the tokens as an indication of focus:

$$\sigma_{\ell,h} = \text{std}(P_{\ell,h,:}, \dim = 1), \tag{10}$$

and select $\delta$ heads, $H_{\text{top}}$, with the lowest standard deviation. The use of standard deviation provides a principled way to quantify how much a head differentiates among tokens. A low standard deviation suggests that eviction scores are nearly uniform, meaning the head lacks a distinct focus and would otherwise reduce to random sampling. In contrast, a higher standard deviation indicates that the head assigns differentiated importance to tokens, reflecting a stronger focus. Finally, for the selected heads, we recalculate the eviction score using expanded observation windows:

$$P_{\ell,h,t} = \frac{1}{O'} \sum_{q=T-O'+1}^{T} A_{\ell,h,q,t}, \quad \text{for } h \in H_{\text{top}}. \tag{11}$$

This modified eviction scores encourage higher coverage compared to the original eviction score. Next, we discuss our cross-layer coverage module that encourages coverage across the layers.

### 4.2 Cross-Layer Coverage

The cross-layer coverage module aims to reduce redundancy and increase the diversity of selected tokens across the layers of the model. Let's define the coverage score of a token $t$ till layer $l$, coverage$_{\ell,t}$, as the number of layers where $t$ has been selected (by at least one head), normalized by the number of layers processed:

$$\text{coverage}_{\ell,t} = \frac{n_t}{l+1}, \tag{12}$$

where $n_t$ is the number of layers in which token $t$ has been selected.

At layer $\ell$, our goal is to prioritize the selection of tokens with lower coverage to increase the overall coverage, while also considering two factors: (1) the global importance of each token, and (2) prioritizing the high-important tokens for that specific layer.

To satisfy the first criterion, we compute the global importance $I_{\ell,t}$ of each token as the maximum attention score over all heads, averaged over a window of size $O$:

$$I_{\ell,t} = \frac{1}{O} \sum_{q=T-O+1}^{T} \max_h A_{\ell,h,q,t}. \tag{13}$$

We then define a focus score that balances importance and novelty (low coverage) as:

$$\text{focus}_{\ell,t} = I_{\ell,t} \cdot (1 - \text{coverage}_{\ell,t}). \tag{14}$$

---

**Algorithm 1** K-VEC: Cross-Head and Cross-Layer Coverage Priority Adjustment

---

1: **Input**: Attention scores $A \in \mathbb{R}^{L \times H \times T \times T}$, $L$ is the total number of layer, $H$ is the number of heads, $T$ is the sequence length, budget $B$, window size $O$, extended window $O'$, number of heads $\delta$, hyperparameter $\lambda$ and $\beta$, current layer $\ell$, coverage counters $n_t$
2: $P_{\ell,h,t} = \frac{1}{O} \sum_{q=T-O+1}^{T} A_{\ell,h,q,t}$                ▷ Initial eviction scores from recent window
3: $\sigma_{\ell,h} = \text{std}(P_{\ell,h,:})$           ▷ Compute variability across tokens for each head
4: $H_{\text{top}} \leftarrow \delta$ heads with lowest $\sigma_{\ell,h}$           ▷ Select most focused heads
5: **for** each head $h \in H_{\text{top}}$ **do**
6:      $P_{\ell,h,t} = \frac{1}{O'} \sum_{q=T-O'+1}^{T} A_{\ell,h,q,t}$           ▷ Recompute using a larger window
7: **end for**
8: $I_{\ell,t} = \frac{1}{O} \sum_{q=T-O+1}^{T} \max_h A_{\ell,h,q,t}$           ▷ Max attention across heads
9: $\text{coverage}_{\ell,t} = \frac{n_t}{\ell+1}$           ▷ Estimate token coverage so far
10: $\text{focus}_{\ell,t} = I_{\ell,t} \cdot (1 - \text{coverage}_{\ell,t})$           ▷ Bias towards less attended tokens
11: $\text{focus}_{\ell,h,t} = \text{expand}(\text{focus}_{\ell,t})$           ▷ Broadcast focus across heads
12: $P'_{\ell,h,t} = P_{\ell,h,t} + \lambda \cdot \text{focus}_{\ell,h,t}$           ▷ Adjust eviction scores with focus bias
13: Identify top-$\beta \cdot B$ tokens in $P_{\ell,h,t}$           ▷ Preserve important tokens
14: Set $P'_{\ell,h,t} = 1.0$ for preserved tokens           ▷ Force high priority for preserved tokens
15: **Return**: $P'_{\ell,h,t}$           ▷ Updated eviction scores

---

The focus score is then broadcast to the head dimension as: $\text{focus}_{\ell,h,t}$. The final priority score is updated using this focus score:

$$P'_{\ell,h,t} = P_{\ell,h,t} + \lambda \cdot \text{focus}_{\ell,h,t}, \tag{15}$$

where $\lambda$ is a hyperparameter controlling the trade-off between original priority and cross-layer coverage.

To ensure that the most critical tokens at the current layer are retained, we select the top $K = \beta \cdot B$ tokens per head based on the original priority $P_{\ell,h,t}$, where $\beta$ is a hyper-parameter. For these top tokens across the heads, we override their scores to ensure they are not evicted:

$$P'_{\ell,h,t} = 1.0, \quad \text{if } t \in \text{top-}K. \tag{16}$$

We present the whole K-VEC algorithm in Algorithm 1.

## 5 Experiments

### 5.1 Experimental setup

**Datasets.** We evaluate K-VEC on 16 subsets of the LongBench dataset (Bai et al., 2023), a comprehensive benchmark for long-context tasks. These subsets span diverse domains, including single-document question answering, multi-document question answering, summarization, few-shot learning, synthetic tasks, and code, with an average sequence length of 6,711 words. The datasets provide a robust testbed for assessing K-VEC's ability to maintain performance under varying KV cache budgets. We use the official evaluation metrics for each subset.

**Base Model.** We use Llama-3.1-8B-Instruct (Grattafiori et al., 2024) as our base model — a widely adopted open-source LLM known for its strong performance on long-context tasks. This model employs Grouped Query Attention (GQA), which reduces the KV cache size to one-quarter of the original compared to standard multi-head attention. Furthermore, as recent KV cache eviction methods have reported results on this encoder, we also adopt it to ensure a fair comparison.

**Baselines.** We compare K-VEC against SOTA KV cache eviction methods, such as SnapKV (Li et al., 2024), PyramidKV (Zhang et al., 2024a), and AdaKV (Feng et al., 2024), which serve as foundational baselines due to their focus on attention-based token selection. We include StreamingLLM (SLM) (Xiao et al., 2023) as a representative sliding window eviction method for reference. Additionally, we report the performance on

Table 1: Detailed results of Llama-3.1-8B-Instruct on LongBench.

| | Single-Doc. QA | | | Multi-Doc. QA | | | Summarization | | | Few-shot Learning | | | Synthetic | | Code | | Ave. Score |
|---|---|---|---|---|---|---|---|---|---|---|---|---|---|---|---|---|---|
| | NrtvQA | Qasper | MF-en | HotpotQA | 2WikiMQA | Musique | GovReport | QMSum | MultiNews | TREC | TriviaQA | SAMSum | PCount | PRe | Lcc | RB-P | |
| Full Cache | 30.22 | 45.37 | 55.80 | 55.97 | 45.00 | 31.26 | 35.12 | 25.38 | 27.20 | 72.50 | 91.64 | 43.57 | 9.41 | 99.50 | 62.88 | 56.43 | 49.20 |
| **B=128** | | | | | | | | | | | | | | | | | |
| SLM | 22.24 | 20.87 | 31.72 | 44.02 | 37.55 | 24.54 | 18.76 | 21.09 | 18.48 | 40.50 | 84.41 | 38.82 | **8.00** | 99.50 | 57.02 | 47.29 | 38.43 |
| Pyramid | 25.70 | 24.69 | 47.74 | 52.87 | 40.57 | 27.23 | 20.02 | 22.38 | 19.74 | 44.50 | 88.81 | 40.30 | 7.22 | **99.50** | 57.25 | 49.90 | 41.78 |
| SnapKV | 25.54 | 24.45 | 48.03 | 53.31 | 40.75 | 28.19 | 20.13 | 22.36 | 19.55 | 45.50 | 89.20 | 40.62 | 6.97 | **99.50** | 58.45 | 49.90 | 42.03 |
| Ada-Pyramid | **27.07** | 25.61 | 49.30 | 53.02 | 41.29 | 27.83 | 20.70 | 23.18 | 20.38 | 51.50 | **90.76** | 40.62 | 6.92 | 99.00 | 59.30 | 50.88 | 42.96 |
| Ada-SnapKV | 24.90 | 24.41 | 49.95 | 53.15 | 41.73 | 28.55 | 20.54 | 23.21 | 20.28 | 50.50 | 89.49 | 40.71 | 7.45 | 99.00 | 58.74 | **52.40** | 42.81 |
| **K-VEC** | 25.96 | **35.96** | **50.40** | **54.79** | **47.14** | **28.75** | **22.35** | **23.47** | **21.88** | **52.89** | 90.35 | **41.25** | 7.85 | **99.50** | **59.35** | 51.25 | **44.57** |
| **B=256** | | | | | | | | | | | | | | | | | |
| SLM | 22.71 | 23.79 | 31.80 | 43.43 | 36.55 | 25.55 | 21.29 | 20.68 | 20.67 | 46.00 | 87.11 | 40.82 | 7.20 | 99.50 | 59.89 | 49.19 | 39.76 |
| Pyramid | 25.53 | 33.15 | 51.44 | 55.03 | 42.42 | 28.62 | 22.57 | 23.37 | 22.33 | 56.50 | 91.19 | 41.28 | 6.97 | 99.50 | 60.36 | 51.18 | 44.47 |
| SnapKV | 26.02 | 32.49 | 51.62 | 54.40 | 42.77 | 28.94 | 22.83 | 23.54 | 22.55 | 53.50 | 91.10 | 40.95 | 7.48 | 99.50 | 60.67 | 53.39 | 44.48 |
| Ada-Pyramid | 25.12 | 35.06 | 52.28 | 54.66 | 41.89 | 28.76 | 23.14 | 23.36 | 22.67 | 63.00 | 90.72 | 41.21 | 7.75 | 99.50 | 61.47 | 53.09 | 45.23 |
| Ada-SnapKV | 26.11 | 33.39 | 51.44 | 54.94 | 42.15 | 29.54 | 23.01 | 23.85 | 22.88 | **63.50** | 91.57 | 40.94 | 8.00 | 99.50 | **61.95** | 54.33 | 45.44 |
| **K-VEC** | **26.33** | **40.38** | **53.53** | **57.39** | **47.14** | **31.24** | **24.35** | **23.94** | **22.95** | 62.74 | **91.89** | **41.95** | **8.05** | 99.50 | 60.87 | **54.90** | **46.70** |
| **B=512** | | | | | | | | | | | | | | | | | |
| SLM | 25.51 | 25.78 | 34.19 | 45.01 | 35.91 | 24.93 | 23.61 | 21.26 | 23.57 | 57.50 | 87.86 | 41.44 | 6.98 | 96.50 | 60.85 | 51.02 | 41.37 |
| Pyramid | 28.71 | 39.89 | 52.86 | 54.00 | 44.20 | 31.22 | 24.74 | 23.73 | 24.28 | 66.00 | 91.07 | 41.42 | 8.39 | 99.50 | 61.99 | 53.44 | 46.59 |
| SnapKV | 29.22 | 40.01 | 53.15 | 54.47 | 43.63 | 31.32 | 25.04 | 23.77 | 24.19 | 64.00 | 92.05 | 41.57 | 8.01 | 99.50 | 63.21 | 55.05 | 46.76 |
| Ada-Pyramid | 28.04 | 40.63 | 53.03 | 54.71 | 43.39 | 30.26 | 25.35 | 24.12 | 24.61 | **69.00** | 91.79 | 42.55 | 7.95 | 99.50 | 62.28 | 54.49 | 46.98 |
| Ada-SnapKV | 29.07 | 40.16 | 52.44 | 53.90 | 43.05 | 31.10 | 25.75 | 24.39 | 24.85 | **69.00** | **92.34** | 42.05 | 7.98 | 99.50 | **63.43** | 55.32 | 47.15 |
| **K-VEC** | **30.04** | **42.81** | **56.07** | **57.90** | **47.39** | **31.39** | **26.24** | **24.86** | **24.92** | 67.47 | 91.62 | **42.67** | **8.45** | 99.50 | 62.90 | **55.68** | **48.12** |
| **B=1024** | | | | | | | | | | | | | | | | | |
| SLM | 24.97 | 30.22 | 37.06 | 46.57 | 39.14 | 25.24 | 26.01 | 21.08 | 25.72 | 63.50 | 88.87 | 42.28 | 6.98 | 89.00 | 61.30 | 53.40 | 42.58 |
| Pyramid | 29.62 | 43.66 | 54.10 | 55.06 | 44.22 | 31.30 | 27.27 | 24.30 | 25.72 | 68.50 | 91.27 | 41.96 | 7.73 | 99.50 | 63.13 | 55.85 | 47.70 |
| SnapKV | 29.28 | 43.64 | 54.34 | 54.24 | 44.34 | 31.52 | 27.80 | 24.39 | 25.95 | 69.00 | **91.72** | 42.50 | 7.80 | 99.50 | 62.99 | 56.45 | 47.84 |
| Ada-Pyramid | 28.76 | 44.57 | 53.73 | 54.89 | 44.15 | **31.97** | 27.75 | **25.26** | 25.84 | 70.50 | 91.62 | 42.37 | 7.67 | 99.50 | 62.96 | 56.52 | 48.00 |
| Ada-SnapKV | 29.23 | 44.09 | 53.82 | 54.80 | 44.01 | 31.40 | 28.86 | 24.73 | **26.04** | **72.50** | **91.72** | 42.56 | 7.82 | 99.50 | **63.22** | 56.33 | 48.16 |
| **K-VEC** | **30.04** | **45.03** | **56.64** | **58.20** | **48.09** | 31.39 | **28.98** | 24.99 | **26.04** | 71.49 | **91.72** | **42.69** | **8.45** | 99.50 | 62.99 | **56.55** | **48.92** |

the full KV cache without eviction. The results on the existing methods are borrowed from the SnapKV (Li et al., 2024) and AdaKV (Feng et al., 2024) papers, and follow their default evaluation settings to evaluate our method.

**Implementation details.** Our proposed K-VEC eviction policy is applied during the pre-fill phase of each layer, following standard practices for KV cache eviction (Li et al., 2024). We evaluate K-VEC across four KV cache budget sizes: 128, 256, 512, and 1024 tokens, reflecting a range of memory constraints. For K-VEC's cross-head coverage module, we select the top $\delta = 3$ heads based on the standard deviation of attention scores, with an extended observation window size of $O' = 32$. The cross-layer coverage module uses $\lambda = 1.0$ to balance token importance and coverage. All other experimental details and parameters are adapted from the SnapKV. All experiments are conducted on an NVIDIA A100 80GB GPU.

## 5.2  Main Results

In Table 1, we present the performance of K-VEC on 16 subsets of LongBench (Bai et al., 2023) using Llama-3.1-8B-Instruct (Grattafiori et al., 2024) for cache sizes $B = 128, 256, 512, 1024$, where lower cache sizes correspond to higher efficiency but may compromise performance. Table 1 shows that performance for existing methods degrades at low cache sizes (e.g., $B = 128$), likely due to suboptimal key-value (KV) cache eviction in existing methods, which discards critical tokens. The drop in performance is more prominent in some of the more context-sensitive tasks, such as Qasper (Dasigi et al., 2021) and TREC (Li & Roth, 2002), where the SOTA method shows 19.76 and 21.0 points drops, respectively.

In contrast, compared to existing methods, K-VEC demonstrates consistent improvements across all tasks and cache sizes. The improvements are especially pronounced at lower cache sizes, with an average performance gain of 1.61 points across the 16 LongBench subsets with the cache budget of 128. For context-sensitive tasks, where existing methods show significant degradation, K-VEC's improvements are even more substantial. For instance, at $B = 128$, K-VEC achieves a 10.35% improvement on Qasper (Dasigi et al., 2021), highlighting its ability to retain critical KV pairs. Similarly, K-VEC outperforms existing SOTA at $B = 256$ by 1.26 points on average, and by up to 5.32 points on individual tasks. A similar trend is observed for the

Table 2: Detailed results of Llama-3.1-8B-Instruct on LongBench and its comparison to HeadKV (Fu et al., 2024).

| | Single-Doc. QA | | | Multi-Doc. QA | | | |
| | NrtvQA | Qasper | MF-en | Hot. | 2Wiki | Musi. | Ave. Score |
|---|---|---|---|---|---|---|---|
| Full Cache | 30.22 | 45.37 | 55.80 | 55.97 | 45.00 | 31.26 | 49.20 |
| | | | B=128 | | | | |
| SnapKV | 22.11 | 15.79 | 31.01 | 41.12 | 29.20 | 19.35 | 26.43 |
| HeadKV-R | 23.49 | 25.39 | 38.15 | 42.45 | 32.84 | 19.95 | 30.38 |
| HeadKV-R2 | 21.80 | 29.19 | 41.89 | 43.73 | 35.01 | 20.40 | 32.00 |
| K-VEC | 24.01 | 34.18 | 42.13 | 44.24 | 39.15 | 21.99 | 34.28 |
| | | | B=1024 | | | | |
| SnapKV | 25.76 | 27.50 | 38.38 | 43.40 | 34.81 | 20.07 | 31.65 |
| HeadKV-R2 | 21.80 | 29.19 | 41.89 | 43.73 | 35.01 | 20.40 | 32.00 |
| HeadKV-R2 | 24.66 | 30.82 | 39.56 | 43.97 | 36.47 | 22.24 | 32.95 |
| K-VEC | 24.75 | 41.77 | 42.99 | 45.75 | 40.24 | 22.45 | 36.33 |

Table 3: Detailed results of Llama-3.1-8B-Instruct on LongBench with cache size of 1024.

| | NrtvQA | Qasper | MF-en | Hot. | 2Wiki | Musi. | GovR. | QM. | Mult. | TREC | Tri. | SAM. | PC. | PRe. | Ave. |
|---|---|---|---|---|---|---|---|---|---|---|---|---|---|---|---|
| SLM | 24.97 | 30.22 | 37.06 | 46.57 | 39.14 | 25.24 | 26.01 | 21.08 | 25.72 | 63.50 | 88.87 | 42.28 | 6.98 | 89.00 | 40.47 |
| Pyramid | 29.62 | 43.66 | 54.10 | 55.06 | 44.22 | 31.30 | 27.27 | 24.30 | 25.68 | 68.50 | 91.27 | 41.96 | 7.73 | 99.50 | 46.01 |
| SnapKV | 29.28 | 43.64 | 54.34 | 54.24 | 44.34 | 31.52 | 27.80 | 24.39 | 25.95 | 69.00 | 91.72 | 42.50 | 7.80 | 99.50 | 46.14 |
| Ada-Pyramid | 28.76 | 44.57 | 53.73 | 54.89 | 44.15 | 31.97 | 27.75 | 25.26 | 25.84 | 70.50 | 91.62 | 42.37 | 7.67 | 99.50 | 46.33 |
| Ada-SnapKV | 29.23 | 44.09 | 53.82 | 54.80 | 44.01 | 31.40 | 28.86 | 24.73 | 26.04 | 72.50 | 91.72 | 42.56 | 7.82 | 99.50 | 46.51 |
| GemFilter | 20.71 | 11.00 | 29.28 | 19.12 | 17.01 | 13.01 | 30.37 | 21.75 | 25.17 | 63.00 | 90.70 | 42.50 | 7.15 | 92.22 | 34.50 |
| K-VEC | 30.04 | 45.03 | 56.64 | 58.20 | 48.09 | 31.39 | 28.98 | 24.99 | 26.04 | 71.49 | 91.72 | 42.69 | 8.45 | 99.50 | 47.38 |

other two settings. Overall, K-VEC's robust performance across diverse tasks and cache budgets validates its effectiveness for efficient long-context processing.

Next, we provide additional evaluations and comparisons to HeadKV (Fu et al., 2024) and GemFilter (Shi et al., 2024) in Tables 2 and 3. We omit these results from the main results table since HeadKV and GemFilter did not report results on all datasets in LongBench. As evident from the results, K-VEC outperforms both HeadKV and GemFilter across the datasets, as well as on average.

### 5.3 Discussion

### 5.3.1 KV-Cache eviction pattern

In this section, we present a qualitative example comparing KV-Cache eviction in SnapKV with our proposed solution. This comparison is visualized in Figure 4. The figure plots the tokens selected after eviction across the layers for an input sequence of length 161. The heatmap values (ranging from 0 to 8) represent the number of attention heads that retain each token. A value of at least 1 indicates that the token is attended to by at least one head in that layer; and any non-zero value in a column implies that the token is retained by at least one layer in the model.

As evident in SnapKV, the eviction policy mostly retains tokens near the end of the sequence and exhibits a clustering pattern in certain sections of the input. Additionally, for most tokens, the selection pattern remains similar across layers, resulting in the appearance of a few prominent horizontal bars. In contrast, the eviction pattern of E-VEC ensures broader coverage by selecting a more diverse set of tokens across layers. While there is some overlap in the selection pattern between K-VEC and SnapKV, due to the retention of some important tokens, the performance boost in our method arises from its emphasis on coverage and the avoidance of repetitive selection across layers and heads. This helps prevent the complete loss of a significant number of tokens, a limitation observed in existing methods.

As we further observe for individual modules, using only the cross-layer coverage module promotes diversity among the selected tokens, as the prominent horizontal bars seen in SnapKV are no longer present. However, this leads to redundancy across heads, indicated by dark dots showing that all heads select the same tokens. In contrast, applying only the cross-head coverage module diversifies token selection across heads, as seen

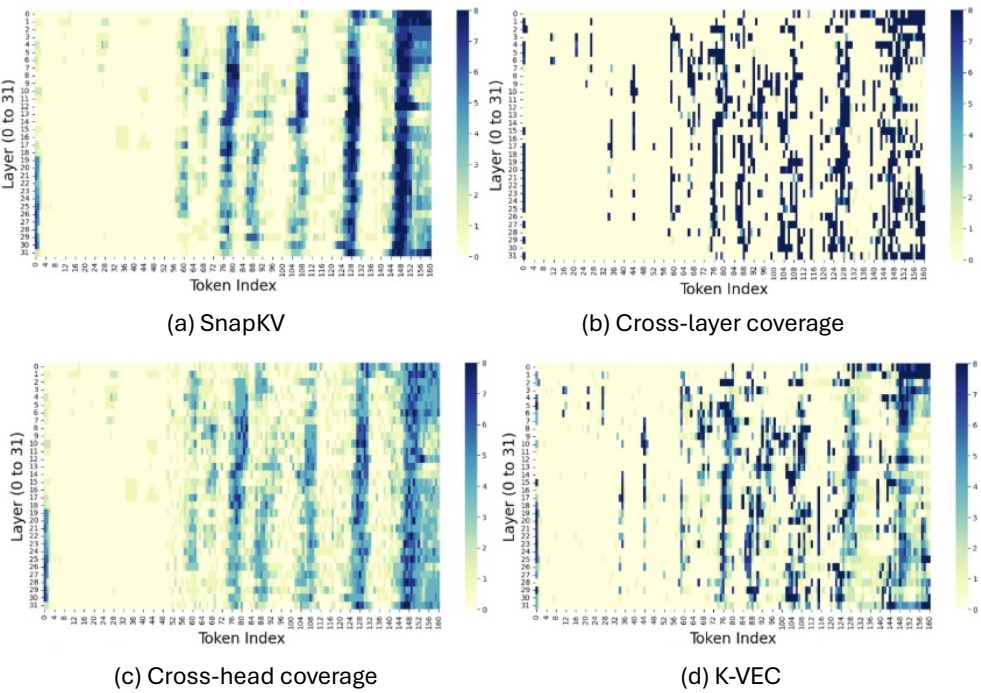

(a) SnapKV

(b) Cross-layer coverage

(c) Cross-head coverage

(d) K-VEC

Figure 4: Visualization of token selection across layers for cross-layer coverage, cross-head coverage and final K-VEC in comparison to SnapKV.

Table 5: Detailed results of Qwen2.5-7B-Instruct on LongBench.

| | Single-Doc. QA | | | Multi-Doc. QA | | | Summarization | | | Few-shot Learning | | | Synthetic | | Code | | |
|---|---|---|---|---|---|---|---|---|---|---|---|---|---|---|---|---|---|
| | NrtvQA | Qasper | MF-en | HotpotQA | 2WikiMQA | Musique | GovReport | QMSum | MultiNews | TREC | TriviaQA | SAMSum | PCount | PRe | Lcc | RB-P | Ave. Score |
| B=128 | | | | | | | | | | | | | | | | | |
| SnapKV | 20.42 | 30.71 | 40.36 | 50.46 | 38.47 | 23.71 | 20.69 | 20.68 | 17.88 | 50.11 | 83.45 | 38.12 | 2.11 | 85.0 | 10.78 | 9.38 | 33.89 |
| **K-VEC** | 23.45 | 34.15 | 39.45 | 51.83 | 40.34 | 26.54 | 22.28 | 23.38 | 18.90 | 49.23 | 85.44 | 41.23 | 2.89 | 84.50 | 11.83 | 10.23 | **35.35** |
| B=1024 | | | | | | | | | | | | | | | | | |
| SnapKV | 23.88 | 38.59 | 46.43 | 55.12 | 42.3 | 27.57 | 27.09 | 23.1 | 24.56 | 53.45 | 84.87 | 39.82 | 2.11 | 84.5 | 10.43 | 8.67 | 37.03 |
| **K-VEC** | 24.55 | 40.02 | 47.23 | 55.98 | 41.49 | 29.22 | 28.63 | 23.79 | 24.78 | 55.73 | 85.99 | 42.57 | 3.01 | 84.91 | 12.19 | 10.95 | **38.19** |

from the broader spread of selected tokens, though visible bars remain due to similar tokens being selected across layers. By combining both strategies in K-VEC, we achieve diversity of selected tokens across both heads and layers.

### 5.3.2 Generalization across LLMs

In this section, we present additional results for the Qwen2.5-7B-Instruct model and its performance compared to SnapKV on two cache sizes: 128 and 1024. The results, detailed in Table 5, show that K-VEC demonstrates a similar performance improvement trend as seen with the Llama-3.1-8BInstruct model. On average, K-VEC provides considerable improvement across both cache sizes.

### 5.3.3 Needle-in-a-Haystack test

Needle-in-a-Haystack is a popular evaluation for testing whether large language models can retrieve and reason over small, specific pieces of information hidden within long contexts. We follow the evaluation

Table 4: Needle-in-a-Haystack test

| Method | Avg. score |
|---|---|
| SnapKV | 87.4 |
| HeadKV | 98.2 |
| K-VEC | 98.2 |

protocol of existing literature (Wu et al., 2024; Fu et al., 2024) and insert the information randomly at different position in the long context. We report the results in Table 4. As evident from this evaluation, K-VEC performs better than SnapKV and is on par with HeadKV in this evaluation.

### 5.3.4 Coverage of tokens

In this section, we analyze the token coverage of our proposed method in comparison to the existing approach. As shown in Table 6, the overall coverage achieved by our method is significantly higher than that of SnapKV. This improvement in coverage also leads to a notable boost in performance. Nonetheless, it is important to note that our coverage-aware eviction strategy does not aim for full coverage, as many tokens inherently contain no relevant information for the given context. Forcing full coverage in the cache will divert the budget from more important tokens, which may lead to a drop in performance.

In Table 7, we present the drop in coverage for different existing methods at different eviction rates. As shown in the table, all existing methods suffer a significant drop in coverage, which—as discussed in the main paper—correlates with a decline in performance. In contrast, our proposed solution demonstrates a considerably smaller drop in coverage.

Table 6: Performance and coverage for our method, in comparison to existing method

| Method | Performance | Coverage |
|--------|-------------|----------|
| SnapKV | 40.01 | 86.6% |
| Ours | 42.81 | 94.5% |

Table 7: Coverage analysis

| Method | 0.25 | 0.5 | 0.75 | 0.9 |
|--------|------|-----|------|-----|
| Pyramid | -3.5% | -4.6% | -8.6% | -20.1% |
| SnapKV | -2.5% | -3.9% | -8.5% | 19.2% |
| Ada-SnapKV | -2.1% | -3.1% | -7.9% | -16.5% |
| K-VEC | -1.1% | -2.5% | -4.5% | -8.6% |

### 5.3.5 Computational Complexity

In this section, we discuss the computational complexity of our proposed method in comparison to SnapKV. Table 8 presents the memory usage and processing time during both the pre-fill and decoding stages of inference. As shown in the table, the memory required to store the KV cache is the same as that of existing methods, since the total number of tokens stored by our approach matches that of prior solutions. Similarly, the decoding speed of our method is equivalent to that of SnapKV. The only notable difference occurs during the pre-fill stage, where our method is slightly slower due

Table 8: Comparison of computational complexity

| | | Tokens/sec | | |
|--------|--------|----------|--------|-------|
| Method | Memory | Pre-fill | Decode | Total |
| SnapKV | 0.1 GB | 5672 | 40.16 | 20.37 |
| Ours | 0.1 GB | 3440 | 40.97 | 18.55 |

to the application of the cross-head and cross-layer coverage strategy. However, since the pre-fill stage is a one-time operation at the beginning of inference, the overall time complexity of our approach remains comparable to that of existing methods. For outputs with long sequences (i.e., a large number of decoding steps), the impact of the slightly slower pre-fill stage becomes negligible, making our method practically as efficient as existing approaches.

Furthermore, a detailed analysis of inference time versus sequence length is presented in Table 9. Specifically, we report the inference time for varying input token lengths and generated output token lengths, and compare the results with SnapKV. Here, the rows represent input token length, and columns represent decoded token length, and the values represent the inference time in seconds. As K-VEC takes slightly longer during the prefill stage, it results in a marginally higher overall inference time when the number of decoded tokens is small. However, as the

Table 9: Comparison of inference time in seconds: SnapKV/K-VEC

| In/Out | 100 | 200 | 300 | 2000 |
|--------|-----|-----|-----|------|
| 500 | 2.52/2.54 | 5.03/5.03 | 7.48/7.47 | 49.88/48.98 |
| 1000 | 2.61/2.68 | 5.15/5.18 | 7.64/7.62 | 49.98/49.99 |
| 2000 | 2.84/3.04 | 5.33/5.48 | 7.86/7.89 | 50.15/49.40 |
| 10000 | 4.26/5.34 | 6.74/7.78 | 9.23/10.22 | 51.56/51.72 |

LLM generates more tokens, the difference between SnapKV and K-VEC diminishes. In modern LLM applications involving long-form reasoning, where output sequences are typically much longer, the additional overhead introduced by K-VEC becomes negligible.

Table 10: Ablation and sensitivity analysis of different components and hyperparameters in K-VEC on the Qasper subset of LongBench (Llama-3.1-8B-Instruct, $B = 256$).

(a) **Main Ablation**

| Configuration | Score |
|---|---|
| Full K-VEC | 35.96 |
| w/o Head Coverage | 34.61 |
| w/o Layer Coverage | 30.87 |
| w/o Both Coverages | 25.54 |

(b) **Observation Window Size**

| $O'$ | Score |
|---|---|
| 20 | 31.58 |
| 24 | 33.24 |
| 32 | 35.96 |
| 48 | 34.56 |

(c) **Layer-wise Cov. Weight**

| $\lambda$ | Score |
|---|---|
| 0.1 | 33.53 |
| 0.5 | 35.40 |
| 1.0 | 35.96 |
| 2.0 | 34.32 |

(d) **Layer-wise Retention Ratio**

| $\beta$ (%) | Score |
|---|---|
| 10 | 34.98 |
| 25 | 35.96 |
| 50 | 35.39 |
| 75 | 35.74 |

(e) **Number of Adjusted Heads**

| $\delta$ | Score |
|---|---|
| 1 | 34.26 |
| 2 | 35.35 |
| 3 | 35.96 |
| 4 | 35.38 |

(f) **Head Selection Criteria**

| Criteria | Score |
|---|---|
| Entropy | 34.17 |
| SD | 35.96 |

## 5.4 Ablation Study

In this section, we present a detailed ablation and parameter sensitivity study on the proposed components of K-VEC. The results of these experiments are summarized in Table 10. Below, we discuss each experiment in detail.

**Main ablation.** Table 10a evaluates the effectiveness of K-VEC's head and layer coverage mechanisms. As shown in the results, removing head coverage reduces the score to 34.61, and removing layer coverage further lowers it to 30.87. When both mechanisms are removed, performance drops significantly to 25.54. In contrast, the full K-VEC configuration achieves the highest score of 35.96, demonstrating that both head and layer coverage contribute substantially to overall performance.

**Observation window size.** Table 10b examines the impact of the observation window size ($O'$), which defines the token span used to assess the importance of entries in the KV cache. The optimal performance, with a score of 35.96, is achieved at $O' = 32$. Smaller window sizes ($O' = 20$ with a score of 31.58 and $O' = 24$ with a score of 33.24) likely miss important contextual information, resulting in suboptimal eviction decisions. Conversely, a larger window ($O' = 48$, score 34.56) may include irrelevant tokens, introducing noise. Thus, an observation window of $O' = 32$ offers the best balance between context coverage and precision.

**Layer-wise coverage weight.** Table 10c analyzes the layer-wise coverage weight ($\lambda$), which balances importance-based eviction and coverage diversity across layers. The best score of 35.96 is obtained with $\lambda = 1.0$. Lower weights ($\lambda = 0.1$, score 33.53) undervalue coverage, which may reduce diversity. Higher weights ($\lambda = 2.0$, score 34.32) slightly degrade performance by over-prioritizing coverage at the expense of importance-based eviction.

**Important token retention ratio.** Table 10d investigates the effect of the layer-wise retention ratio of important tokens, $\beta$, which determines the fraction of tokens retained based on the original eviction score alone. The highest performance score of 35.96 is achieved with a 25% retention ratio. A lower ratio (10%, score 34.98) retains too few tokens based on importance and puts excessive focus on coverage. While higher ratios (50%, score 35.39; 75%, score 35.74) reduce the selection of coverage-driven tokens. The 25% ratio offers an optimal balance between coverage and the preservation of important tokens.

**Number of adjusted heads.** Table 10e explores the impact of the number of adjusted attention heads ($\delta$), where K-VEC applies the cross-head coverage strategy. The highest score of 35.96 is achieved with $\delta = 3$. Lower values ($\delta = 1$, score 34.26; $\delta = 2$, score 35.35) provide insufficient coverage, while higher values may alter the contribution of important heads, leading to suboptimal performance. The peak at $\delta = 3$ reflects a balanced trade-off between coverage and the preservation of token importance.

**Head Selection Criteria.** In Table 10f, we explore different criteria for selecting the head to enforce coverage. Specifically, we explore entropy and standard deviation as the selection criteria. As shown, the standard deviation serves as a better indication of less focused heads, and enforcing head-wise coverage.

# 6 Conclusion

In this work, we propose K-VEC, a coverage-aware KV-cache eviction strategy that addresses the memory and scalability challenges of deploying large language models in long-context applications. By introducing cross-head and cross-layer coverage modules, K-VEC ensures the retention of critical tokens, significantly improving performance over existing methods, particularly at low cache budgets. Our evaluations on the LongBench dataset demonstrate K-VEC's superiority across diverse tasks, with up to 10.35 points gains at a 128-token budget. Ablation and sensitivity analyses further confirm the robustness and effectiveness of our proposed modules. K-VEC offers a practical solution for efficient LLM inference, enabling scalable deployment in real-time and resource-constrained environments.

**Limitations.** As discussed in the computational complexity analysis, one limitation of our proposed solution is a slight increase in compute cost during the pre-fill stage. However, this overhead diminishes for long-generation tasks, as the decoding speed remains comparable to existing methods. In modern LLM applications involving long-form reasoning, where output sequences are typically much longer, the additional overhead introduced by K-VEC becomes negligible. Additionally, while our approach improves coverage, it does not achieve full coverage. Nonetheless, full coverage is not necessarily beneficial, as some tokens may not contribute meaningful information in the context of the query. Future work could explore strategies to further increase coverage and investigate integration with other optimization techniques to enhance the overall efficiency of LLMs.

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
