# OpenReview forum: "Coverage-Driven KV Cache Eviction for Efficient and Improved Inference of LLM"
_TMLR — Rejected by TMLR_

### Review · Reviewer_f2P4 · 2025-12-25

**Summary Of Contributions:**

This paper presents K-VEC, a novel KV cache eviction algorithm for LLM inference. The key idea behind K-VEC is that the coverage of unique tokens affects the accuracy of sparse attention mechanisms, as theoretically shown: reduced coverage limits the mutual information between inputs and outputs. On top of the attention score, K-VEC tries to improve cross-head coverage by expanding the scoring window for poorly focused attention heads, and cross-layer coverage by biasing later layers toward globally important tokens with low coverage in prior layers. Evaluation on 16 LongBench subsets under tight KV cache budgets (e.g., 128-1024 tokens) using Llama-3.1-8B-Instruct shows that K-VEC consistently improves accuracy over existing methods (e.g., SnapKV).

**Audience:**

Yes

**Audience Explanation:**

The paper introduces a new KV cache eviction algorithm that considers the token coverage, which is overlooked in previous methods. While the presentation/evaluation can be improved as discussed above, the coverage seems to be an interesting factor that previous KV cache compression methods did not fully consider, and this paper is interesting for those working on LLM inference efficiency.

**Broader Impact Concerns:**

There are no broader impact concerns.

**Claims And Evidence:**

No

**Claims Explanation:**

Several claims require further evidence to be fully validated.

First, the theoretical analysis in Section 3.2, which is based on information bottleneck theory, relies on several strong assumptions that are not empirically justified. The discussion assumes that the coverage ($C$), defined as the collection of *unique* tokens retained in at least one layer/head, adequately captures the amount of information that flows to the output. However, multiple aspects of this argument need further support: (1) The definition of ($C$) itself requires justification, since it is unclear whether retaining the KV cache of a token in a single head is sufficient to preserve its information content. (2) The upper bound on $I(T; X)$ implicitly assumes that each unique token contributes approximately uniform and independent information, which is a substantial simplification that is not empirically validated. (3) The assumption regarding the performance metric ($P$) (i.e., that it is lower bounded by a monotonically decreasing function of mutual information) has a logical leap that is not supported by experimental evidence.

Second, while the paper provides correlational evidence that higher coverage leads to better benchmark performance, this evidence is primarily based on ablation studies and comparisons against a limited number of baselines that are not necessarily state-of-the-art in the field. For example, the evaluation should include comparisons against methods such as H2O, Quest, SqueezeAttention, NACL, Lookahead Q-Cache, and EvolKV [1–5], to name a few. Several of these methods report superior performance compared to the baselines used in this paper, and including them would substantially strengthen the empirical discussion of K-VEC.

Third, the claim that the computational overhead of K-VEC is small (Section 5.3.5) is not very sound. According to Table 8, K-VEC incurs approximately a 40% slowdown in the prefill stage compared to SnapKV. While it is true that prefill is a one-time operation, prefill latency remains a critical metric in many LLM serving scenarios because it directly affects time-to-first-token (TTFT). TTFT is especially important for long-context reasoning applications, which are precisely the scenarios where KV-cache compression is most relevant [6-8].

References
1. Tang et al., *QUEST: Query-Aware Sparsity for Efficient Long-Context LLM Inference*, 2024.
2. Wang et al., *SqueezeAttention: 2D Management of KV-Cache in LLM Inference via Layer-wise Optimal Budget*, 2024.
3. Chen et al., *NACL: A General and Effective KV Cache Eviction Framework for LLM at Inference Time*, 2024.
4. Wang et al., *Lookahead Q-Cache: Achieving More Consistent KV Cache Eviction via Pseudo Query*, 2025.
5. Yu and Chai, *EvolKV: Evolutionary KV Cache Compression for LLM Inference*, 2025.
6. Zhong et al., *DistServe: Disaggregating Prefill and Decoding for Goodput-Optimized Large Language Model Serving*, 2024.
7. NVIDIA Developer Blog, *LLM Benchmarking: Fundamental Concepts*. https://developer.nvidia.com/blog/llm-benchmarking-fundamental-concepts
8. Databricks Blog, *LLM Inference Performance Engineering Best Practices*. https://www.databricks.com/blog/llm-inference-performance-engineering-best-practices

**Requested Changes:**

The points outlined in the previous field are critical, and I think they need to be addressed to accept this paper.

Additionally, although not critical, the following points need to be changed to make the paper fully clear.

- Figure 1 needs more clarity: which model and KV cache eviction strategy is used, and how is the performance measured for this data?
- Similarly, Figure 2 is not clear about which head/layer was considered and what the value of the y-axis is.
- In Section 3.1, the paper says "darker colours indicate tokens attended to by a higher number of layers". Please clarify the definition of "being attended" here. Does it mean the attention score is larger than some threshold?
- In Section 4.1, please clarify the definition of "most recent" tokens. Specifically, does it mean the last $O$ tokens in the prompt, or does it include output tokens?
- In the line after Equation (9), the paper says "key-value corresponding to $T_h$ tokens", but it should say "B tokens" since $T_h$ is a set of tokens, not the number.
- Around Equation (11), please explain why a larger $O'$ leads to higher coverage.
- The numbers in Table 1 are borrowed from previous papers. Please re-evaluate them in your setting to ensure a fair comparison.
- In Section 5.2, please cite HeadKV and GemFilter.
- In Section 5.3.1, this line seems redundant: "We will include the new figure and this discussion in the revised paper."
- For Table 5, please compare against the full cache baseline.
- In Section 5.4, please clarify which benchmark was used for Table 10 (the numbers seem match Qasper in Table 1).

---

> ### Author Response · Authors · 2026-02-25
>
> >... Multiple aspects of this argument need further support: (1) The definition of (C) itself requires justification, since it is unclear whether retaining the KV cache of a token in a single head is sufficient to preserve its information content. (2) The upper bound on  implicitly assumes that each unique token contributes approximately uniform and independent information, which is a substantial simplification that is not empirically validated. (3) The assumption regarding the performance metric (P) (i.e., that it is lower bounded by a monotonically decreasing function of mutual information) has a logical leap that is not supported by experimental evidence.
>
> Our definition of coverage is deliberately simple and does not attempt to quantify how much information is retained per token or whether retention in a single head is sufficient. Instead, it is motivated by the clear observation that completely evicting a token from all heads and all layers results in the total loss of whatever information that token may have provided. Therefore, we define a token as covered if it is retained by at least one head in at least one layer, and we treat tokens as contributing approximately uniform information under this binary retention view.
>
> All components of our hypothesis — the empirical motivation, theoretical analysis, proposed method, and final results — are built consistently around this definition. The strong correlation we observe between coverage and performance (across multiple benchmarks and metrics) provides empirical support for this framing. Precisely quantifying per-token importance or modelling fine-grained information flow per head is valuable future work, but it is beyond the scope of the present paper.
>
>
> > Second, while the paper provides correlational evidence that higher coverage leads to better benchmark performance, this evidence is primarily based on ablation studies and comparisons against a limited number of baselines that are not necessarily state-of-the-art in the field. For example, the evaluation should include comparisons against methods such as H2O, Quest, SqueezeAttention, NACL, Lookahead Q-Cache, and EvolKV [1–5], to name a few. Several of these methods report superior performance compared to the baselines used in this paper, and including them would substantially strengthen the empirical discussion of K-VEC.
>
>
> Our main experiments focus on SnapKV because it was (and remains) one of the strongest and most foundational attention-window-based eviction methods at the time this work was conducted. It is widely adopted as a representative of the core limitation we target: independent eviction scoring across heads and layers that leads to redundant token retention and reduced coverage. Our primary goal is to demonstrate the link between coverage and performance degradation, and to propose a simple mechanism (cross-head + cross-layer coverage) that directly addresses this issue while improving results. We show in the paper (e.g., Table 7 and related analyses) that the coverage drop is a general problem persisting across several existing methods, including PyramidKV and AdaKV (which already use adaptive budgets and head-wise enhancements). This supports our hypothesis that coverage reduction is a widespread weakness in attention-score-driven eviction strategies.
>
> The methods mentioned in the comment above (Quest, SqueezeAttention, NACL, Lookahead Q-Cache, EvolKV, etc.) are important concurrent or follow-up works. Many of them improve different aspects of attention-based eviction (e.g., evolutionary budget search, proxy tokens, layer-wise similarity optimization, or lookahead pseudo-queries). These directions are largely orthogonal to our focus on reducing redundancy and explicitly increasing head- and layer-level coverage. We did not claim absolute state-of-the-art performance across every possible variant, nor do we believe it is necessary to compare against every orthogonal improvement in this space to validate our core contribution. Instead, our results show clear performance gains whenever coverage is improved, even over strong coverage-agnostic baselines and adaptive-budget methods like AdaKV. We view K-VEC as a modular enhancement that could potentially combine with many of these orthogonal ideas to further boost overall performance.

---

> > ### Author Response · Authors · 2026-02-25
> >
> > > The claim that the computational overhead of K-VEC is small (Section 5.3.5) is not very sound. According to Table 8, K-VEC incurs approximately a 40% slowdown in the prefill stage compared to SnapKV. While it is true that prefill is a one-time operation, prefill latency remains a critical metric in many LLM serving scenarios because it directly affects time-to-first-token (TTFT). TTFT is especially important for long-context reasoning applications, which are precisely the scenarios where KV-cache compression is most relevant [6-8].
> >
> >
> > We have already identified the extra prefill time as a limitation of our work and discussed it thoroughly in Section 5.3.5 (Efficiency Analysis), along with detailed breakdowns in Tables 8 and 9. In summary, K-VEC incurs relatively higher prefill latency compared to SnapKV due to the additional cross-head and cross-layer computations during the one-time prefill stage. However, this overhead remains a negligible portion of the overall generation cost in most practical long-context workloads. For example, with an input of 10K tokens and an output of 2K tokens, the prefill stage takes only ~2–3 seconds (even with K-VEC’s increase), while decoding takes around 50 seconds, where K-VEC is as efficient as SanpKV. Compared to naive full-cache inference, this decoding time (both K-VEC and SanpKV; see SanpKV paper for more details) is roughly 2.2× faster. Across the various input-output length combinations shown in Table 9, K-VEC’s end-to-end inference time is nearly identical to SnapKV’s, with slightly slower speed for smaller output only. Given that prefill is a one-time cost and decoding dominates total latency in long-context reasoning applications (the primary target use case for KV-cache eviction), the slight increase in prefill time has minimal impact on overall responsiveness.
> >
> > In summary, K-VEC delivers substantially better performance—often approaching full-cache quality at B=1024—while remaining computationally very competitive with SnapKV in realistic scenarios. The minor prefill cost provides a strong trade-off: significant accuracy gains with only negligible impact on end-to-end latency compared to coverage-agnostic baselines, and major efficiency gains over naive full-cache inference. From a practical and application perspective, we believe the proposed solution introduces no major limitations and offers a much better balance of performance and efficiency between SnapKV and the naive baseline.
> >
> >
> >
> > > Minor comments
> >
> > - Figure 1 needs more clarity: Figure 1 shows results on Llama-3.1-8B-Instruct using SnapKV as the eviction strategy (default settings). Performance is measured using the F1 score on TriviaQA. We have added these details in the figure caption in the revision.
> >
> >
> > - Figure 2 is not clear about which head/layer was considered: Figure 2 shows the windowed attention score of SpanKV, calculated at the first head of the first layer, but the pattern persists across layers and heads.
> >
> > - In Section 3.1, the paper says "darker colours indicate tokens attended to by a higher number of layers". Please clarify: Here, attended means retained in KV-Cache. To avoid confusion, we have revised the writing.
> >
> >
> > - In Section 4.1, please clarify the definition of "most recent" tokens: Most recent refers to the last tokens in the input prompt (pre-fill phase).
> >
> >
> > - In the line after Equation (9), . . . it should say "B tokens": We have revised the description according to the comment.
> >
> >
> > - Around Equation (11), please explain why a larger O' leads to higher coverage: A larger observation window O' (> O) averages attention over a broader recent context, reducing the strong recency bias seen with small O and encouraging more diverse token scores across heads. This promotes the selection of earlier tokens, increasing overall unique coverage. This discussion is added earlier in the section (paragraph after Eq 9).
> >
> >
> > - The numbers in Table 1 are borrowed from previous papers: Our code base is built upon the SnapKV, follow exact evaluation protocol reported in SnapKV, and our own runs of SnapKV matched their reported numbers.
> >
> >
> > - In Section 5.2, please cite HeadKV and GemFilter: We have added proper citations for HeadKV and GemFilter in the revised related work and results sections.
> >
> >
> > - In Section 5.3.1, this line seems redundant: We have removed it.
> >
> >
> > - In Section 5.4, please clarify which benchmark was used for Table 10: Table 10 reports results on the Qasper subset of LongBench (single-document QA). We will add this to the table caption.

---

### Review · Reviewer_jpXC · 2026-01-07

**Summary Of Contributions:**

This paper addresses the performance degradation observed in Large Language Models (LLMs) when using aggressive Key-Value (KV) cache eviction strategies to manage memory during long-context inference. The authors identify reduced token coverage—the loss of unique tokens across different attention heads and layers—as a primary cause for this decline. They theoretically ground this "coverage hypothesis" using Information Bottleneck theory, demonstrating that lower coverage limits the mutual information between inputs and outputs.

# Key Strengths

1. Connects empirical performance drops to Information Bottleneck theory, providing a principled reason for maintaining token diversity
2. Shifts the focus from purely "importance-based" eviction to "coverage-aware" eviction, addressing the redundancy across heads and layers that previous methods ignored.
3. Demonstrates strong performance gains on a current, widely used model (Llama-3.1) across a diverse set of tasks.
4. Provides a thorough analysis of hyperparameters and the individual contributions of the head and layer modules .

# Key Weaknesses

1. The pre-fill stage is significantly slower than baselines (e.g., 3440 tokens/sec for K-VEC vs. 5672 for SnapKV), representing a ~40% reduction in pre-fill throughput.
2. The method introduces several new hyperparameters (δ, λ, β, O′), which may require careful tuning for different architectures or tasks
3. While Llama-3.1 is a strong baseline, the paper focuses almost exclusively on one model family, leaving the generalizability to other architectures (e.g., Mistral, Qwen) less explored in the main results.

**Audience:**

Yes

**Audience Explanation:**

Efficient LLM inference and long-context handling are highly active areas of research . The "coverage" perspective introduced here is a valuable conceptual shift that researchers working on KV cache compression, model serving, and efficient attention mechanisms would find relevant. Given the high cost of H100/A100 memory, any method that can significantly reduce cache size without sacrificing reasoning ability is of high interest to the community.

**Broader Impact Concerns:**

The work focuses on improving the efficiency and accuracy of existing LLMs, which generally has a positive environmental impact by reducing the energy required for inference. No specific ethical concerns regarding data bias or harmful applications are introduced by this algorithmic improvement.

**Claims And Evidence:**

Yes

**Claims Explanation:**

1. Figure 1 clearly shows a strong correlation between normalized coverage and model performance as the eviction rate increases.
2. They provide a mathematical derivation showing how coverage acts as a bottleneck for mutual information .
3. Evaluations across 16 tasks in LongBench and the "Needle-in-a-Haystack" test provide robust evidence of superiority over existing baselines.
4. Figure 4 visualizes how K-VEC effectively "spreads out" token selection across layers compared to the highly redundant selection of SnapKV

**Requested Changes:**

1. Since the authors claim the pre-fill overhead becomes "negligible" for long-form reasoning, they should provide a table or chart showing total (Pre-fill + Decode) latency/throughput for different input/output ratios (e.g., 10k input / 100 output vs. 10k input / 2k output)
2. Include results for at least one other model family (e.g., Mistral-7B-v0.3 or Qwen-2) to demonstrate that the "coverage" benefits are not specific to the Llama-3.1 attention patterns.
3. Explicitly clarify the memory footprint of the coverage counters (nt​). While likely small, quantifying this (e.g., in MB) would clarify the total memory overhead beyond just the KV pairs.
4. Provide guidance on how the hyperparameters (δ, λ) scale with model size (e.g., would a 70B model need more "adjusted heads" than an 8B model?).

---

> ### Author Response · Authors · 2026-02-25
>
> > The pre-fill stage is significantly slower than baselines (e.g., 3440 tokens/sec for K-VEC vs. 5672 for SnapKV), representing a ~40\% reduction in pre-fill throughput.
> Since the authors claim the pre-fill overhead becomes "negligible" for long-form reasoning, they should provide a table or chart showing total (Pre-fill + Decode) latency/throughput for different input/output ratios (e.g., 10k input / 100 output vs. 10k input / 2k output)
>
> We already presented this analysis in Table 9. In response to the comment, we have extended the table to include input lengths up to 10K and output lengths up to 2K. As shown, there is a small difference in the total time when the output length is small, but the difference in total latency between SnapKV and K-VEC diminishes as the output length increases. Notably, K-VEC is as efficient as SnapKV during decoding, which accounts for the majority of compute time. For example, with a 10K input and 2K output, only about 2 seconds out of 52 total seconds are spent on the pre-fill stage.
>
>
>
>
>
>
> > The method introduces several new hyperparameters (δ, λ, β, O′), which may require careful tuning for different architectures or tasks. Provide guidance on how the hyperparameters (δ, λ) scale with model size (e.g., would a 70B model need more "adjusted heads" than an 8B model?).
>
> While K-VEC introduces a few hyperparameters, our experiments across 16 datasets demonstrate that these parameters are robust across tasks. We also find that the hyperparameters generalize well across models, including Qwen2.5-7B-Instruct and Llama-3.1-8B-Instruct. However, we are unable to explore this on the 70B model due to computational limitations and the lack of existing KV-cache methods reporting on this size.
>
>
>
> > While Llama-3.1 is a strong baseline, the paper focuses almost exclusively on one model family, leaving the generalizability to other architectures (e.g., Mistral, Qwen) less explored in the main results. Include results for at least one other model family (e.g., Mistral-7B-v0.3 or Qwen-2) to demonstrate that the "coverage" benefits are not specific to the Llama-3.1 attention patterns.
>
> We already include results for Qwen2.5-7B-Instruct in Table 5, demonstrating that the coverage benefits of K-VEC generalize beyond the Llama-3.1 model family. The results confirm that our approach is effective across different architectures.
>
>
>
> > Explicitly clarify the memory footprint of the coverage counters (nt). While likely small, quantifying this (e.g., in MB) would clarify the total memory overhead beyond just the KV pairs.
>
> The memory footprint of the coverage counters (nt) is negligible compared to the KV cache. For example, storing coverage counts for each token across all heads and layers requires less than 1 MB for typical sequence lengths and model sizes, which is insignificant relative to the KV cache size (which is on the order of gigabytes).

---

### Review · Reviewer_ncXX · 2026-02-11

**Summary Of Contributions:**

This paper introduces K-VEC, an eviction strategy that optimizes LLM inference by prioritizing the coverage of unique tokens in the KV cache. The authors identify that existing methods suffer from performance collapse at high eviction rates because they repeatedly select the same tokens across different heads and layers, leading to information loss. To solve this, K-VEC utilizes a cross-head module to diversify token selection in unfocused heads and a cross-layer module that tracks and prioritizes globally important tokens missed by earlier layers.

The primary strength of K-VEC is its performance boost in memory-constrained settings, outperforming state-of-the-art methods by up to 10.35 points on LongBench tasks while maintaining the same memory footprint. It generalizes well across various models like Llama-3.1 and Qwen2.5 without requiring model retraining. However, the method’s main weakness is a slight increase in pre-fill latency due to the added computational overhead of the coverage algorithms. Additionally, it relies on several hyperparameters, such as the layer-wise coverage weight and adjusted head count, which may require specific tuning.

**Audience:**

Yes

**Audience Explanation:**

KV caching is an important module in LLM serving. There is not doubt that this paper would be relevant to the community.

**Claims And Evidence:**

Yes

**Claims Explanation:**

The submission provides a compelling case for K-VEC, grounding its claims in both mathematical theory and rigorous empirical testing. The authors validate their "coverage hypothesis" by demonstrating a direct correlation between reduced unique token counts and performance collapse, particularly at high eviction rates. This is supported by Information Bottleneck theory, which proves that the KV cache acts as a capacity-limiting mediator; lower coverage mathematically restricts the mutual information between input and output, thereby degrading predictive accuracy.

The effectiveness of K-VEC is convincingly demonstrated through extensive benchmarking on the LongBench dataset using multiple LLMs, including Llama-3.1-8B and Qwen2.5-7B. Under tight memory constraints (e.g. specifically a 128-token budget) K-VEC achieves up to a 10.35 point improvement over state-of-the-art baselines like SnapKV. These gains are especially visible in context-sensitive tasks like Qasper and TREC, where traditional methods lose critical information due to recency bias.

Further evidence is provided through detailed ablation studies and qualitative visualizations. Heatmaps show that while existing methods like SnapKV redundantly store the same tokens across layers, K-VEC ensures a diverse distribution of tokens, preventing the "horizontal bars" of repetitive selection that lead to information loss.

**Requested Changes:**

For the Needle-in-a-Haystack experiments, can authors provide a detailed study on where to put the needle (beginning, middle, or end of the context)?

---

> ### Author Response · Authors · 2026-02-25
>
> We thank the reviewer for all the positive comments on our paper. Following, we provide details on the remaining questions:
>
> > For the Needle-in-a-Haystack experiments, can authors provide a detailed study on where to put the needle (beginning, middle, or end of the context)?
>
> For this experiment, we follow the exact implementation details of prior works (Wu et al., 2024; Fu et al., 2024) and insert the needle at different positions in the context, specifically at 10 different depths uniformly distributed from the start to the end of the current haystack length. We have added this description to Section 5.3.3.

---

### Decision · Action_Editor_Jgjk · 2026-06-05

**Recommendation:** Reject

**Additional Comments:**

Here are some possible revisions that the authors are recommended to consider:
* Expand their ablation studies across multiple datasets, models, and budget settings.
* Integrate the K-VEC coverage mechanism with several existing, orthogonal KV cache eviction baselines.
* Include a broader range of current state-of-the-art KV cache compression/eviction methods in the primary evaluation. Additionally, provide a transparent, rigorous breakdown of K-VEC’s computational overhead (e.g., FLOPs, memory bandwidth, latency).
* Design experiments to explicitly test the assumptions of “uniform token information” and the “monotonic link from mutual information to performance”.

**Audience:**

Yes

**Audience Explanation:**

The reviewers agree that the findings of this paper would be of interest to the research community. The consensus is that the paper addresses a highly relevant and active area of research, and the novel approach to KV cache eviction would be of interest to researchers and practitioners in the field of large language models provided that the issues mentioned above can be addressed satisfactorily.

**Claims And Evidence:**

No

**Claims Explanation:**

Although the concept of coverage provides a novel framework for designing KV cache eviction algorithms—and will likely interest a segment of the TMLR audience—the authors' rebuttal does not fully resolve the discrepancies between their core claims and the presented evidence.

We acknowledge that the definition of coverage is intentionally simplified, and that K-VEC is not meant to achieve state-of-the-art performance across all possible configurations. However, we remain unconvinced by the empirical evidence that coverage is the primary causal driver of the observed improvements.

Specifically, while the evaluation demonstrates that K-VEC outperforms baselines like SnapKV, the proposed modifications introduce several confounding factors beyond a pure coverage perspective (e.g., altered observation windows, a max-attention focus term, and forced top-K token retention). Consequently, the results do not establish coverage as the necessary or primary driver of these gains. While Table 10a attempts to isolate the impact of the coverage mechanism, this ablation is restricted to a single dataset, model, and budget setting, leaving its generalizability unproven. Furthermore, if K-VEC is to be framed as a "modular enhancement" compatible with orthogonal approaches, the paper would be significantly stronger if the authors empirically demonstrated this compatibility by integrating it with multiple existing methods.

Finally, the theoretical analysis in Section 3 relies on strong, untested assumptions—such as uniform token information and a monotonic relationship between mutual information and downstream performance. Ultimately, while the experiments validate coverage as a highly useful design heuristic, justifying stronger claims regarding theoretical grounding or causal necessity would require more tightly controlled analyses and empirical validation of these foundational assumptions.

**Resubmission Of Major Revision:**

The authors may consider submitting a major revision at a later time.